# Dating the Emergence of Human Endemic Coronaviruses

**DOI:** 10.3390/v14051095

**Published:** 2022-05-19

**Authors:** Diego Forni, Rachele Cagliani, Uberto Pozzoli, Alessandra Mozzi, Federica Arrigoni, Luca De Gioia, Mario Clerici, Manuela Sironi

**Affiliations:** 1Bioinformatics, Scientific Institute IRCCS E. MEDEA, 23842 Bosisio Parini, Italy; rachele.cagliani@lanostrafamiglia.it (R.C.); uberto.pozzoli@lanostrafamiglia.it (U.P.); alessandra.mozzi@lanostrafamiglia.it (A.M.); manuela.sironi@lanostrafamiglia.it (M.S.); 2Department of Biotechnology and Biosciences, University of Milan-Bicocca, 20126 Milan, Italy; federica.arrigoni@unimib.it (F.A.); luca.degioia@unimib.it (L.D.G.); 3Department of Physiopathology and Transplantation, University of Milan, 20122 Milan, Italy; mario.clerici@unimi.it; 4Don Carlo Gnocchi Foundation ONLUS, IRCCS, 20148 Milan, Italy

**Keywords:** coronavirus, molecular dating, emerging infectious diseases, viral disease, virus evolution

## Abstract

Four endemic coronaviruses infect humans and cause mild symptoms. Because previous analyses were based on a limited number of sequences and did not control for effects that affect molecular dating, we re-assessed the timing of endemic coronavirus emergence. After controlling for recombination, selective pressure, and molecular clock model, we obtained similar tMRCA (time to the most recent common ancestor) estimates for the four coronaviruses, ranging from 72 (HCoV-229E) to 54 (HCoV-NL63) years ago. The split times of HCoV-229E and HCoV-OC43 from camel alphacoronavirus and bovine coronavirus were dated ~268 and ~99 years ago. The split times of HCoV-HKU1 and HCoV-NL63 could not be calculated, as their zoonoticic sources are unknown. To compare the timing of coronavirus emergence to that of another respiratory virus, we recorded the occurrence of influenza pandemics since 1500. Although there is no clear relationship between pandemic occurrence and human population size, the frequency of influenza pandemics seems to intensify starting around 1700, which corresponds with the initial phase of exponential increase of human population and to the emergence of HCoV-229E. The frequency of flu pandemics in the 19th century also suggests that the concurrence of HCoV-OC43 emergence and the Russian flu pandemic may be due to chance.

## 1. Introduction

Coronaviruses (order *Nidovirales*, family *Coronaviridae*, subfamily *Coronavirinae*) are a diverse group of positive-sense, single-stranded RNA enveloped viruses with high zoonotic potential [1,2,3]. In 2002, a highly pathogenic coronavirus, severe acute respiratory syndrome coronavirus (SARS-CoV), spilled over from palm civets to humans and caused ~8000 cases in several countries [4]. These events were followed by the appearance, in 2012, of Middle East respiratory syndrome coronavirus (MERS-CoV), a camel-derived pathogen that caused multiple outbreaks of respiratory disease mainly in the Arabic Peninsula [5]. Containment and surveillance strategies allowed for the control of these viruses, which have never (SARS-CoV) or only occasionally (MERS-CoV) reappeared in human populations [6,7]. However, at the end of 2019, SARS-CoV-2 emerged in China and is now recognized as the cause of COVID-19 [8]. The virus rapidly spread worldwide, and the World Health Organization declared the SARS-CoV-2 pandemic in early March 2020.

The epidemic behavior of SARS-CoV, MERS-CoV, and SARS-CoV-2, as well as their clinical severity, have clearly raised awareness of the potential danger posed by coronaviruses, which were considered relatively harmless to humans before 2002. In fact, four other human coronaviruses (HCoV) (HCoV-OC43, HCoV-HKU1, HCoV-NL63, and HCoV-229E), sometimes referred to as “common cold coronaviruses”, have been circulating in human populations for decades, usually causing mild symptoms [2,9,10].

Like the highly pathogenic coronaviruses, the endemic coronaviruses have a zoonotic origin [2,3,11]. Although with some controversy [12], most previous estimates indicated that the endemic coronaviruses entered human populations in the last 1000 years [2,13,14,15,16,17,18]. However, these early analyses were often based on a small number of sequences and did not control for effects that are now recognized to affect molecular dating (e.g., recombination). For instance, using the first complete genome of HCoV-OC43 and the sequences of 15 BCoV spike proteins, Vijgen and coworkers estimated that the time to the most recent common ancestor (tMRCA) of HCoV-OC43 and BCoV dated to the end of the 19th century [14]. A follow-up analysis with seven additional HCoV-OC43 sequences and based on both the S and N sequences confirmed this estimate, placing the tMRCA at the end of the 19th century or at the beginning of the 20th [17]. A study that sequenced and analyzed several BCoV spike sequences also reached the same conclusion [18]. With respect to HCoV-229E, previous dating analysis did not include the camelid viruses, as they were unavailable at that time [13,15]. In the wake of the COVID-19 pandemic, a better understanding of the evolutionary dynamics of endemic coronaviruses, as well as of the tendency of viral disease emergence, might provide valuable insight into the possible trajectories of SARS-CoV-2 evolution.

## 2. Materials and Methods

### 2.1. Sequences and Alignments

Complete or almost complete genome sequences for all four endemic coronaviruses were downloaded from the NCBI database (National Center for Biotechnology Information, http://www.ncbi.nlm.nih.gov/, accessed on 15 April 2022). Only sequences with known sampling dates were included in the analyses (Appendix A). The HCoV-OC43 Paris strain was excluded, as its sampling date is uncertain [19]. For HCoV-229E and HCoV-OC43, the closest phylogenetically related animal viruses were also retrieved, namely camel alphacoronavirus and bovine coronavirus (BCoV) (Appendix A).

Sequence alignments were generated using MAFFT (v7.427) (multiple alignment using fast Fourier transform) [20], with default parameters.

#### Recombination Analysis

Recombination can affect phylogenetic tree branch length estimates and, consequently, molecular evolution analyses [21]. Thus, each coronavirus alignment was tested for evidence of recombination signals using the 3SEQ software (v.1.7, software for identifying recombination in sequence data) [22]. This method scans a given alignment searching for mosaic recombination signals in all possible sequence triplets. The result is the identification of genomic regions in which one of the three sequences is the recombinant (child) of the other two (parental). 3SEQ full scans were run with a recombination significance threshold of 0.01. All significant recombination events were mapped onto coronavirus alignments, and the longest non-recombinant genomic regions, defined as the genomic region between two recombination breakpoints, were selected for subsequent analyses (Figure 1). This generated four non-recombinant alignments with the following lengths: HCoV-OC43: 12,691 nucleotides, HCoV-229E: 17,271 nucleotides, HCoV-HKU1: 10,820 nucleotides, HCoV-NL62: 2320 nucleotides. The unique recombination events identified in genome alignments are shown in Figure 1.

### 2.2. Phylogenetic Trees and Temporal Signal

Phylogenetic trees for the non-recombinant regions of all endemic coronaviruses were reconstructed using the phyML (phylogenetic tree estimation under the maximum likelihood (ML) principle) software under a general time-reversible (GTR) model plus gamma-distributed rates and 4 substitution rate categories [23]. The substitution models were estimated using JmodelTest 2 [24,25].

Internal-GTR-estimated branch lengths were compared to branch lengths calculated using a model that accounts for different selective pressures among lineages. This model is implemented in the aBSREL (adaptive branch site random effects likelihood [26]) tool from the HYPHY (hypothesis testing using phylogenesis) suite (version 2.5) [27].

To evaluate whether the non-recombinant genomic regions selected for the analyses carried sufficient temporal signal, we calculated the correlation coefficients (r) of regressions of root-to-tip genetic distances against sequence sampling years. We applied a previously proposed method, described by Murray and co-workers [28], that minimizes the residual mean squares of the models rather than one that maximizes r^2^ [28]. We calculated *p* values by performing clustered permutations (1000) of the sampling dates, as previously suggested [28,29]. We considered significant a regression with *p* < 0.05 (Figure 1).

### 2.3. Molecular Dating

A time estimate phylogenetic reconstruction was performed using a Bayesian approach implemented in the Bayesian Evolutionary Analysis by Sampling Trees (BEAST, v.1.10.4) software [30].

To select the best-fit molecular clock and tree prior, we ran the path sampling tool implemented in BEAST to choose between a constant size, an exponential growth, or a coalescent Bayesian skyline tree prior, and between a strict and an uncorrelated relaxed log-normal clock (100 steps, 1,000,000 iterations each). For all parameters, default priors were chosen only if they had proper distributions; otherwise, they were changed accordingly (i.e., for the population size parameter, an uninformative lognormal prior distribution was used, instead of the 1/x default prior).

A Bayes factor test was applied to compare the different likelihoods (Appendix A). Since none of the models were favored when compared to all the others, we selected the simplest among the favored ones (Appendix A). Thus, a constant population size tree prior with a strict clock model was used for HCoV-229E and HCoV-NL63, whereas a constant population and a relaxed clock with a log-normal distribution were used for HCoV-OC43.

For the HCoV-HKU1 phylogeny, we used the mean rates estimated for the other betacoronavirus HCoV-OC43 (1.78 × 10^−4^ substitutions per site yr^−1^) as an informative rate prior following a normal distribution.

We performed two different Markov chain Monte Carlo runs for all four endemic coronaviruses, one hundred million iterations each, and sampled every 10,000 steps after a 10% burn-in. The runs were combined after checking for convergence and for heaving effective sampling sizes > 100.

We generated a maximum clade credibility tree using TreeAnnotator [31], which was visualized with FigTree (http://tree.bio.ed.ac.uk/, accessed on 15 April 2022).

### 2.4. Data on Influenza Pandemics and Human Population Size

The timing of influenza pandemics was obtained from a previous work [32], as well as from references therein [33,34,35,36,37,38,39,40,41,42]. Estimates of human population size were obtained from the “Our World in Data” website (https://ourworldindata.org/, accessed on 15 April 2022).

## 3. Results

### 3.1. Time-Frame of Human Endemic Coronavirus Emergence

As mentioned above, all endemic coronaviruses were estimated to have recently emerged as human pathogens [2,13,14,15,16,17,18]. However, besides being generally based on a limited number of sequences, most previous analyses did not include some of the viruses that are now recognized to be closely related to endemic human coronaviruses (e.g., the dromedary camel alphacoronaviruses related to HCoV-229E). Moreover, it is now recognized that the presence of recombination, the lack of a temporal signal in the sequence data, and the pervasive effect of purifying selection can affect molecular dating [21,43,44]. Accounting for these effects has become common practice only in recent years. Thus, because their zoonotic source is known, we decided to reassess the timing of HCoV-OC43 and HCoV-229E emergence. As their animal origin is unknown, we instead estimated the time when circulating strains of HCoV-NL63 and HCoV-HKU1 last shared a common ancestor. For this purpose, we retrieved all available sequences with known sampling date for the four coronaviruses (HCoV-OC43, n = 167; HCoV-229E, n = 31; HCoV-NL63, n = 68; HCoV-HKU1, n = 34), together with the sequences of the reference genomes of BCoV and camel alphacoronavirus (Appendix A).

Because recombination is known to be frequent in all coronavirus genera [45,46,47], we used 3SEQ to identify recombination events, which were detected in all datasets (Figure 1) [22]. Based on the location of breakpoint positions, we then selected the longest non-recombining region for each alignment. Specifically, we obtained relatively long regions for HCoV-OC43, HCoV-229E, and HCoV-HKU1, whereas the non-recombining region was shorter for HCoV-NL63 (Figure 1). For all the selected non-recombining regions, maximum likelihood phylogenetic trees were constructed, and we checked for the presence of a temporal signal by performing regression of root-to-tip genetic distances against sampling dates. These analyses indicated a strong temporal signal for all regions, with the exclusion of the HCoV-HKU1 region (Figure 1). In this latter case, the lack of a temporal signal is most likely due to the short time span among virus sampling dates, with the earliest sequences collected in 2003 (Appendix A).

Before performing molecular dating, we evaluated whether natural selection strongly affected branch length estimates in the viral phylogenies. In fact, it is now recognized that purifying selection and saturation effects contribute to the time-dependent substitution rate variation in viruses, which, in turn, affects molecular dating [43,48]. We thus estimated branch lengths using the aBSREL (adaptive branch-site random effects likelihood) model, which accounts for different selective pressures among lineages and is relatively robust to substitution saturation [49]. For all phylogenies, branch lengths estimated with aBSREL were comparable to those obtained with a GTR (general time reversible) model (Figure 1), suggesting that molecular dating can be performed with minor effects related to the time dependency of substitution rates.

Thus, for the three phylogenies (HCoV-OC43, HCoV-229E, and HCoV-NL63) showing a temporal signal, we used a Bayesian approach to estimate substitution rates and time-measured evolutionary histories. Substitution rates in the range of 1.78 × 10^−4^ to 2.03 × 10^−4^ substitutions per site yr^−1^ were obtained, in line with previous analyses [46]. For the HCoV-HKU1 phylogeny, date estimates were obtained by using the substitution rate of HCoV-OC43 (another betacoronavirus) as a prior. For the circulating strains of all coronaviruses, we obtained similar tMRCA estimates, which ranged from 72 (HCoV-229E) to 54 (HCoV-NL63) years ago (Figure 2 and Appendix A).

The splits of HCoV-OC43 and HCoV-229E from their most closely related animal viruses were more variable. Specifically, we estimated that HCoV-OC43 split from the bovine coronavirus (BCoV) lineage around 1923 (HPD: 1872–1967), whereas HCoV-229E separated from the camel alphacoronavirus in the 18th century (1754, HPD: 1714–1791) (Figure 2 and Appendix A). It should, however, be noted that the 95%HPD intervals for the split of HCoV-OC43 from the animal virus were very large, and the inference should, therefore, be taken with caution.

### 3.2. Human Coronavirus Emergence in the Context of Viral Outbreaks

The molecular dating analyses reported above indicate that, most likely, all endemic coronaviruses emerged as human pathogens earlier than 50 years ago, and possibly in a more distant past (Figure 3). This implies that, at least between ~1970 and 2002 (when SARS-CoV appeared), no coronavirus gained the ability to spread widely in our species. Thus, the pattern of coronavirus emergence seems to be highly irregular and to have intensified in recent years. To compare the timing of coronavirus emergence to that of another respiratory virus, we recorded the occurrence of known influenza pandemics since 1500. As previously noted [50], this pattern is also irregular. Although there is no clear relationship between pandemic occurrence and human population size, the frequency of influenza pandemics seems to intensify starting around 1700, which corresponds with the initial phase of the exponential increase of human population (Figure 3). This time also roughly corresponds to the emergence of HCoV-229E.

## 4. Discussion

Many uncertainties surround the origin of coronaviruses as human pathogens. Recent molecular dating analyses have estimated that the tMRCA of sarbecoviruses dates back to about 21,000 years ago [51], which roughly corresponds to the time when a set of human proteins that interact with coronaviruses started to experience positive selection in Asian populations [52]. Whether the selective pressure was accounted for by a coronavirus or another pathogen remains to be clarified. In the case that the agent that infected humans back then was indeed a betacoronavirus, it must have gone extinct, at least in human populations. Indeed, it seems difficult to imagine that a newly emerged coronavirus might be able to spread and persist in early human communities, which were small and poorly connected. Whatever the nature of that early infectious agent, it is clear that much information on the possible trajectories of SARS-CoV-2 evolution can be gained by the analysis of previous human epidemics, especially those caused by coronaviruses. We thus leveraged the increasing availability of sequence data and advances in molecular dating approaches to re-estimate the time when endemic coronaviruses entered human populations.

Unfortunately, we could not estimate the emergence time for HCoV-HKU1 and HCoV-NL63, as the hosts from which they spilled over are unknown. Both viruses have their closest relatives in wild rodents and bats, but there is no information concerning the unsampled diversity in these mammals or in other domestic ones. Thus, for these two viruses, we can only estimate the time when circulating strains last shared a common ancestor, which was in the 1960s. Thus, we can place an upper-bound limit and state that they emerged earlier than ~50 years ago. This time roughly corresponds to the tMRCAs of HCoV-229E and HCoV-OC43. However, for these viruses, we estimated that the splits from the viruses hosted by domestic animals occurred in the 18th and 20th centuries, respectively. Because bovines and camels are plausible zoonotic sources for human infections, these split dates may be considered as good proxies for the time when HCoV-OC43 and HCoV-229E entered human populations. The long time spans that, especially for HCoV-229E, separate the split times and the tMRCAs are most likely accounted for either by extinct ancestral lineages or by unsampled viral diversity. In the case of HCoV-OC43, our estimate of the split from BCoV in 1923 (HPD: 1872–1967) is based on a large non-recombining region of ORF1a. This result is in good agreement with previous works that analyzed the *S* or *N* gene regions and obtained split dates ranging from 1873 to 1910 (plus credible intervals) [14,17,18]. Previous studies on HCoV-229E, as well as on HCoV-NL63, mainly analyzed the split times from bat viruses and are thus not comparable with the data we present herein [13,15]. In general, our results and previous studies agree that the four endemic coronaviruses entered human populations earlier than 50 years ago [14,15,16,17,18]. This indicates that, for several decades, no coronavirus spilled over to humans (or at least caused a registered outbreak) until three highly pathogenic coronaviruses emerged in tight temporal succession. The factors responsible for the timing of epidemics or pandemics have remained unknown for decades in the case of influenza, as virological and non-virological elements have been associated with the occurrence of pandemics with poor explanatory power [32,34,53,54,55,56,57]. Thus, it is presently impossible to predict the timing of such events and their severity. As noted elsewhere [50], there is no clear role of human population size in the frequency of pandemics either. Thus, the recent exponential growth has not determined a comparable increase in pandemic occurrence. However, some intensification seems to be detectable starting around the beginning of the 18th century, which also corresponds to the time when HCoV-229E emerged. This period was characterized by the industrial revolution and the colonial expansion, which resulted in larger cities and long-distance travels. Whether these changes in human behavior contributed to the intensification of viral disease emergence remains to be evaluated. Alternatively, it is possible that the increase in pandemic occurrence since 1700 simply reflects the increasing accuracy and reliability of medical or historical records.

In the same way as we are unable to predict the timing, we have very little ability to anticipate which viruses will emerge and how pathogenic they will be [58,59]. In this respect, it is worth mentioning that, because they have now circulated in (and adapted to) human populations for decades, if not centuries, we cannot exclude that the endemic coronaviruses were once more pathogenic than they are now. Indeed, it was previously suggested that the 1889–1890 flu pandemic (known as the Russian flu), which was characterized by pronounced central nervous system symptoms, was actually caused by HCoV-OC43 [14,60]. If we allow for credible intervals, our estimate of the timing of HCoV-OC43 emergence is still compatible with the hypothesis that the virus, which displays some neurotropism, was the causative agent of Russian flu. However, the frequency of flu pandemics in the 19th century suggests the concurrence of HCoV-OC43 emergence and the Russian flu pandemic may be due to chance. Only the retrieval of historical samples from the pandemic will prove or refute this hypothesis.

## Figures and Tables

**Figure 1 viruses-14-01095-f001:**
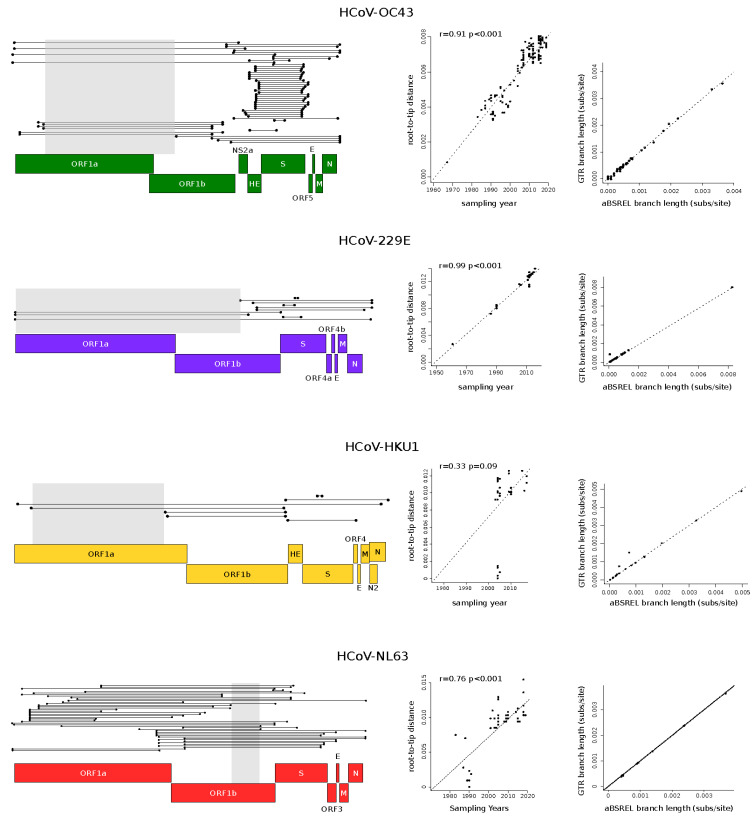
Recombination events and temporal signal. Unique recombination events in endemic coronaviruses (**left panels**). Each event is shown as a line with dots representing the start and the end. The non-recombinant regions used in the analyses are indicated with gray shadows. Schematic representations of coronavirus genomes are also reported. Plots of the root-to-tip distance as a function of sampling years are shown in the central panels. Each point corresponds to a viral sequence and the dotted line is the linear regression calculated using a method that minimizes the residual mean squares. The r coefficient and the corresponding *p* value are also shown. (**right panels**) report comparisons of branch lengths obtained using the aBSREL and the GTR models. Each dot represents an internal branch of a phylogenetic tree calculated using the longest non-recombinant regions of each endemic coronavirus.

**Figure 2 viruses-14-01095-f002:**
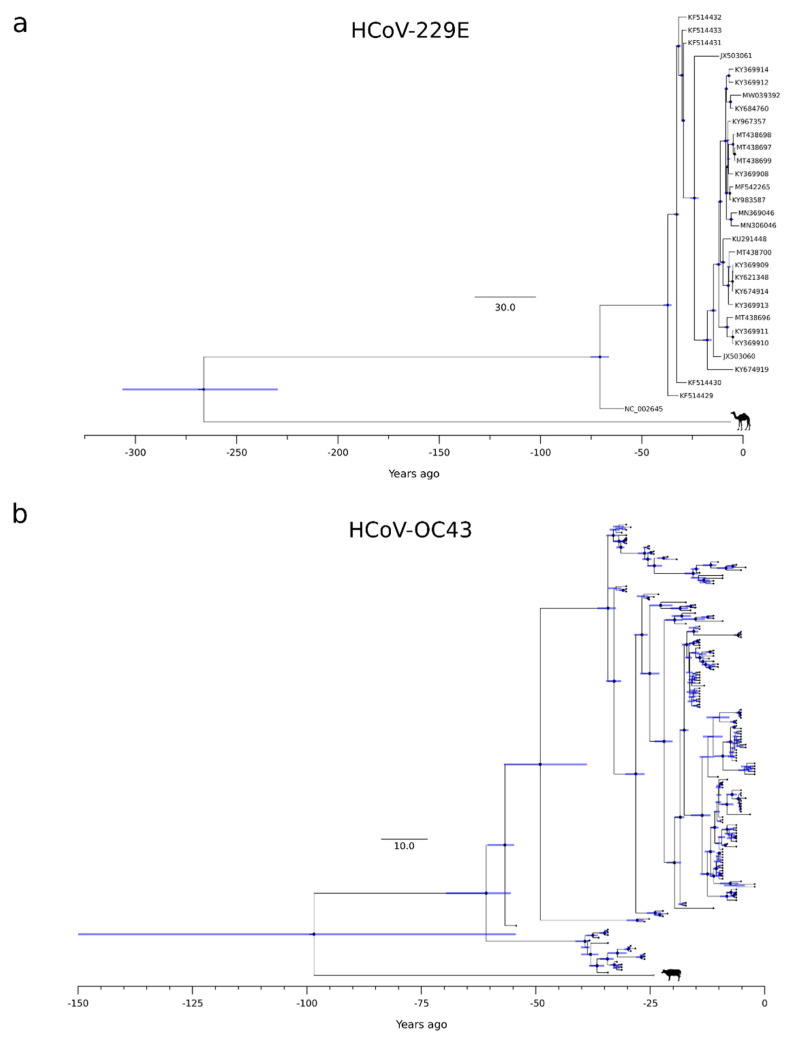
HCoV-229E and HCoV-OC43 timescaled phylogenetic trees. Maximum likelihood trees estimated for the non-recombinant region of HCoV-229E (**a**) and HCoV-OC43 (**b**). Branch lengths represent the evolutionary time measured by the grids corresponding to the timescale shown at the tree base (in years). For internal nodes, 95% HPD bars are shown, and black dots indicate a posterior probability > 0.80 for that node.

**Figure 3 viruses-14-01095-f003:**
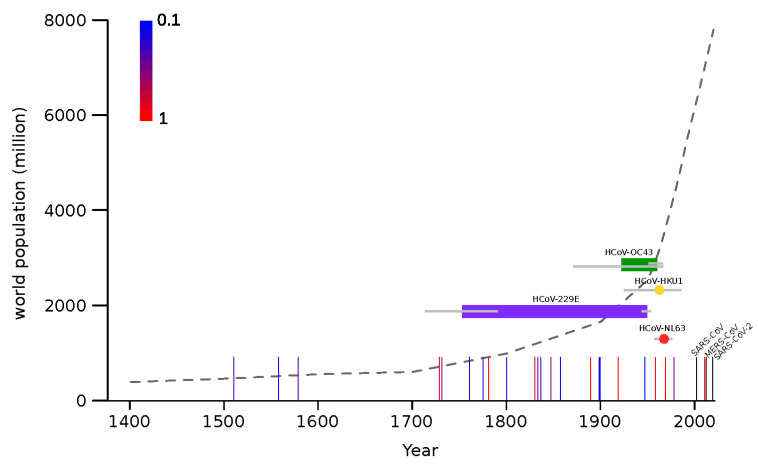
Timeline of endemic coronavirus emergence. Colored horizontal bars represent the time span between the divergence of each coronavirus from the closest known animal virus and the tMRCA (time to the most recent common ancestor) of circulating strains. In the case of HCoV-HKU1 and HCoV-NL63 tMRCA are shown as colored dots. Gray bars indicate 95% HPD. Vertical lines represent influenza pandemic events and the scaled colors (from blue to red, see legend on the plot) indicate the proportion of articles reporting that event as pandemic [33,34,35,36,37,38,39,40,41,42]. The three recent coronavirus zoonoses are shown as black vertical lines. The world human population count from 1400 C.E. is reported as a dotted gray line.

## Data Availability

Lists of virus accession IDs are reported in Appendix A.

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
