# Peer review of "Dating the Emergence of Human Endemic Coronaviruses"

_viruses, 2022, doi:10.3390/v14051095_

Round 1

Reviewer 1 Report

Line 29

I would prefer to add " Enveloped Viruses" to the definition of Coronaviruses.

All abbreviations are to be written in full when they first appear in text

Line 43- Human Coronavirus - OC 43

           - Human Coronavirus - HKU1

           - Human Coronavirus - 229 E

           - Human Coronavirus - NL63

Line 63 MAFFT (Multiple Alignment using Fast Fourier Transform)

Line 67 3SEQ ( 3`- end Sequencing for Expression Quantification)

Line 89 phyML ( Phylogenetic Tree estimation under the Maximum

                         Likelihood (ML) principle

Line 95 HYPHY ( HYpothesis testing using PHYlogenesis

Author Response

Comments and Suggestions for Authors

>>> We really thank the Reviewer for the time he/she dedicated to the assessment of our manuscript and for valuable suggestion.

Line 29

I would prefer to add " Enveloped Viruses" to the definition of Coronaviruses.

>>> Changed as suggested

All abbreviations are to be written in full when they first appear in text

Line 43- Human Coronavirus - OC 43

           - Human Coronavirus - HKU1

           - Human Coronavirus - 229 E

           - Human Coronavirus – NL63

>>> Changed as suggested

Line 63 MAFFT (Multiple Alignment using Fast Fourier Transform)

>>> Changed as suggested

Line 67 3SEQ ( 3`- end Sequencing for Expression Quantification)

>>> The one reported above is not the full name for 3SEQ. Indeed, 3SEQ is named after its working procedure. Thus, the full name was included as follows: “Software For Identifying Recombination In Sequence Data”

Line 89 phyML ( Phylogenetic Tree estimation under the Maximum Likelihood (ML) principle

>>> Changed as suggested

Line 95 HYPHY ( HYpothesis testing using PHYlogenesis

>>> Changed as suggested

Reviewer 2 Report

The authors present a study looking into the emergence of endemic human coronaviruses in the human population. The data is well analyzed and presented and is mostly to the point. I have highlighted some points below for further consideration.

Introduction

Aside from indicating potential shortcomings of previous studies that sought to provide estimates of the emergence of human coronaviruses, an introduction of these studies should also be provided in this section (such as those of Vijgen et al on HCoV-OC43), more so as they are later discussed in comparison with the results from this study.

Methods

The authors describe how temporal signal in the data was estimated by regression analysis and a method that minimizes residual mean squares of models. The authors should mention what software was used for this and if the method for minimizing residual mean squares is an established one, should also be mentioned, otherwise better described to enable replication.

Was there consideration for the different substitution rates in codon positions 1+2 vs 3 with respect to the site model used for analysis? If so this should be stated.

The parameters of the priors used should be shown in a table form.

Results

Portions of the figure 2b are not very legible and should be improved. Particularly the areas with short branch lengths. This will allow a reader to have a better estimation of the HPDs for instance.

Discussion

The authors should compare their estimated spilt times to previous estimates from other studies and discuss.

Author Response

The authors present a study looking into the emergence of endemic human coronaviruses in the human population. The data is well analyzed and presented and is mostly to the point. I have highlighted some points below for further consideration.

>>> We really thank the Reviewer for the time he/she dedicated to the assessment of our manuscript and for valuable suggestion.

Introduction

Aside from indicating potential shortcomings of previous studies that sought to provide estimates of the emergence of human coronaviruses, an introduction of these studies should also be provided in this section (such as those of Vijgen et al on HCoV-OC43), more so as they are later discussed in comparison with the results from this study.

>>> Thank you for this observation. The following sentences were added in the introduction

For instance, using the first complete genome of HCoV-OC43 and the sequences of 15 BCoV spike proteins, Vijgen and coworkers estimated that the time to the most recent common ancestor (tMRCA) of HCoV-OC43 and BCoV dated to the end of the 19th century [14]. A follow-up analysis with 7 additional HCoV-OC43 sequences and based on both the S and N sequences confirmed this estimate, placing the tMRCA at the end of the 19th century or at the beginning of the 20th [17]. A study that sequenced and analyzed several BCoV spike sequences also reached the same conclusion [18]. With respect to HCoV-229E, previous dating analysis did not include the camelid viruses, as they were unavailable at that time [13, 15].”

Methods

The authors describe how temporal signal in the data was estimated by regression analysis and a method that minimizes residual mean squares of models. The authors should mention what software was used for this and if the method for minimizing residual mean squares is an established one, should also be mentioned, otherwise better described to enable replication.

>>> Thank you for this comment. For regression analysis and the minimization of residual mean squares, we used a method that was fully described by Murray and co-workers (doi.org/10.1111/2041-210X.12466). The method is available in the form of R scripts. The methods were modified as follows: “To evaluate whether the non-recombinant genomic regions selected for the analyses carried sufficient temporal signal, we calculated the correlation coefficients (r) of regressions of root-to-tip genetic distances against sequence sampling years [28]. We applied a previously proposed method, described by Murray and co-workers [28], that minimizes the residual mean squares of the models rather than one that maximizes r2 [28]. We calculated p values by performing clustered permutations (1,000) of the sampling dates, as previously suggested [28,29]. We considered significant a regression with p<0.05 (Fig. 1).”

Was there consideration for the different substitution rates in codon positions 1+2 vs 3 with respect to the site model used for analysis? If so this should be stated.

>>> We confirm there was no differentiation of codon positions

The parameters of the priors used should be shown in a table form.

>>> For each CoV dataset, we performed analyses using six models: constant size, exponential growth, or coalescent Bayesian skyline with strict and uncorrelated relaxed log normal clock. This would amount to a large number of tables to be included. Moreover, we used default priors in all cases, excluding when they had improper distributions. Thus, the following sentence was included in the methods: “For all parameters default priors were chosen only if they had proper distributions, otherwise they were changed accordingly (i.e for the population size parameter an uninformative lognormal prior distribution was used, instead of the 1/x default prior).”

Results

Portions of the figure 2b are not very legible and should be improved. Particularly the areas with short branch lengths. This will allow a reader to have a better estimation of the HPDs for instance.

>>> Figure 2 was re-drown to improve readability. Please note that it is however difficult to have full visibility of HPDs for very short branches.

Discussion

The authors should compare their estimated spilt times to previous estimates from other studies and discuss.

>>> Thank you for this suggestion. The following lines were added in the discussion: “In the case of HCoV-OC43, our estimate of the split from BCoV in 1923 (HPD: 1872-1967) is based on a large non-recombining region of ORF1a. This result is in good agreement with previous works that analyzed the S or N gene regions and obtained split dates ranging from 1873 to 1910 (plus credible intervals) [14,17,18]. Previous studies on HCoV-229E, as well as on HCoV-NL63, mainly analyzed the split times from bat viruses and are thus not comparable with the data we present herein [13,15]. In general, our results and previous studies agree that the four endemic coronaviruses entered human populations earlier than 50 years ago [14-18].”